

# A comparison between predetermined and self-selected approaches in resistance training: effects on power performance and psychological outcomes among elite youth athletes

Kevin Watson[1,2,*], Israel Halperin[3,4,*], Joan Aguilera-Castells[5] and Antonio Dello Iacono[2]

[1] Strength and Conditioning Department, Glasgow School of Sport, Glasgow, United Kingdom
[2] School of Health and Life Sciences, University of the West of Scotland, Glasgow, United Kingdom
[3] School of Public Health, Tel Aviv University, Tel Aviv, Israel
[4] Sylvan Adams Institute, Tel Aviv, Israel
[5] Faculty of Psychology, Education Sciences and Sport, Universitat Rámon Llull, Barcelona, Spain
[*] These authors contributed equally to this work.

Corresponding author
Antonio Dello Iacono,
antonio.delloiacono@uws.ac.uk

## ABSTRACT

**Background.** The aim of this study was to investigate if choice over resistance training exercise order affects motor performance and psychological outcomes among elite youth hockey players.

**Methods.** Seventeen elite hockey players (male, $n = 14$; female, $n = 3$, age: $15.1 \pm 1.1$ years) participated in this study. In the first session, individual optimum power loads were calculated in the back squat, jump squat, bench press and bench throw exercises. Then, in four counterbalanced sessions, participants completed three sets of six repetitions in the same exercises loaded with their optimum power loads. In two sessions, athletes used a self-selected order of exercises, while in other two sessions the order was predetermined. Power outputs were estimated with a linear position transducer. Fatigue and enjoyment were measured during and after the sessions using standardized questionnaires. Repeated measures analyses of variance and a paired-sample $t$-test were used to compare the effects between conditions.

**Results.** We observed trivial to small differences between conditions in power outputs ($p \geq 0.07$; $ES \leq 0.21$), fatigue ($p \geq 0.42$; $ES \leq 0.33$) and enjoyment ($p = 0.72$; $ES = 0.05$).

**Conclusion.** Given the comparable effects between approaches, both can be used when coaching youth athletes. Self-selecting the order of exercises based on preferences is a feasible and practical coaching option when working with youth athletes.

## INTRODUCTION

Designing effective resistance training programs requires careful consideration of how to manipulate the different variables such as training volume and intensity, exercises selection and exercise order. Regarding the latter variable, research has shown that the order of exercises can directly affect subsequent strength adaptations (*Nunes et al., 2020*; *Simao et al., 2012*). Exercises completed early in a session can be executed with higher external loads and superior efficacy due to the minimal concurrent and detrimental effects of local fatigue (*Nunes et al., 2020*; *Simao et al., 2012*). The superior peak and total force associated with exercises performed at the beginning of a session can lead to greater gains in strength capabilities compared to programs in which the same exercises are performed later (*Nunes et al., 2020*; *Simao et al., 2012*). While most studies have focused on force production capabilities, given the intimate relationship between force production and power development (*Enoka & Duchateau, 2008*), exercise order can be expected to influence the latter in a similar fashion. Consequently, resistance training programs are traditionally designed with the exercises prescribed in a fixed and predetermined order before sessions initiation and depending on the main training goals (*Ratamess et al., 2009*; *Sands, Wurth & Hewit, 2012*). Despite the clear benefits of the predetermined approach, implementing it with large groups of athletes, in confined spaces, or with limited equipment, may lead to a number of difficulties. Mainly, athletes need to wait until a piece of equipment is free which may lead to boredom, cooling down, and inefficient usage of time, all of which compromise the efficacy of the session.

A possible alternative to the predetermined approach is the self-selected approach, in which athletes are allowed to make choices concerning various resistance training variables. Choices can include the selection of exercises (*Rauch et al., 2020*), their order (*Nunes et al., 2020*; *Simao et al., 2012*) and number of repetitions completed (*Emanuel et al., 2020*). A growing number of studies report superior or comparable acute performance effects (*Halperin et al., 2017*; *Iwatsuki et al., 2017*) and long-term adaptations (*Colquhoun et al., 2017*; *McNamara & Stearne, 2010*; *Rauch et al., 2020*) following self-selected protocols compared to the predetermined approaches. In addition to effective performance outcomes, the use of self-selected approaches in resistance training has many benefits. First, allowing athletes to choose the order of exercises can overcome logistical challenges associated with coaching large groups with limited equipment. For example, rather than having all athletes wait in line to complete the same exercise on the single piece of available equipment, athletes can begin the session using different exercises. Second, having the freedom to make choices enhances motor learning (*Wulf & Lewthwaite, 2016*) motivation to perform (*Deci & Ryan, 2000*; *Wulf & Lewthwaite, 2016*), perception of well-being (*Adie, Duda & Ntoumanis, 2012*) and the likelihood of adhering to a program (*Silva et al., 2011*). Hence, granting athletes choice in exercise order has unique benefits that may, at times, outweigh the physiological benefits associated with a predetermined exercise order.

In a recent review, *Halperin et al. (2018)* suggested that when adopting autonomy-supportive strategies to plan strength and conditioning sessions, coaches should consider the type, number, and range of choices, along with contextual factors pertaining to the

**Table 1  Anthropometric characteristics of the participants.** Skinfolds: triceps, subscapular, biceps, iliac crest, supraspinale, abdominal, thigh, calf. Skinfolds were collected following the guidelines of International Society for the Advancement of Kinanthropometry.

|  | Male ($n = 14$) | Female ($n = 3$) |
| --- | --- | --- |
| Age (years) | $15.3 \pm 1.3$ | $15.0 \pm 1.0$ |
| Height (cm) | $174.2 \pm 7.7$ | $166.3 \pm 0.2$ |
| Body mass (kg) | $63.2 \pm 10.2$ | $58.0 \pm 1.9$ |
| Sum of eight skinfolds (mm) | $73.1 \pm 34.9$ | $107.1 \pm 21.3$ |

training environment and the coach-athlete relationship. While more studies comparing the effects of self-selected and predetermined approaches in strength and conditioning settings are being carried out (*Colquhoun et al., 2017*; *Emanuel et al., 2020*; *McNamara & Stearne, 2010*; *Rauch et al., 2020*), to our knowledge none have been conducted on youth athletes. Accordingly, the aim of this study is to investigate the effects of choice provision over exercise order in strength and conditioning sessions on acute power performance outcomes and subjective responses of fatigue and enjoyment among youth athletes. Our main hypothesis was that granting athletes with choice would enhance power performance outcomes, increase athletes' enjoyment and reduce perceptions of fatigue.

## MATERIALS & METHODS

### Subjects

The planned sample size consisted of twenty elite academy hockey players. However, fourteen participants (male, $n = 11$ and female $n = 3$) performed all four experimental sessions, while three participants (male, $n = 3$) completed only two sessions, one for each condition (See Table 1 for participants' characteristics). Two participants dropped out due to injuries sustained during sport-specific activities and one participant withdrew before completion of the study. Participants had at least 2 years (range: 2–9) experience of high-level training in a youth academy and at least 1-year (range: 1–6) experience of strength and conditioning training, especially in the four exercises used in this study. However, athletes were mostly familiarized with the predetermined approach, and had little to no experience with the self-selected approach prior to the time of the study commencement. Written informed consent was obtained from the participants or their legal representative after receiving a written and oral explanation of the purpose, benefits and potential risks of the study. All procedures were conducted in accordance with the Helsinki Declaration and approved by the ethics committee of the University of the West of Scotland (IRB: 2019-8652).

### Design

A randomized crossover design (Fig. 1) was used to compare the power outputs and subjective responses of fatigue and enjoyment during four resistance training sessions implementing upper and lower body exercises: bench press, bench throw, back squat and
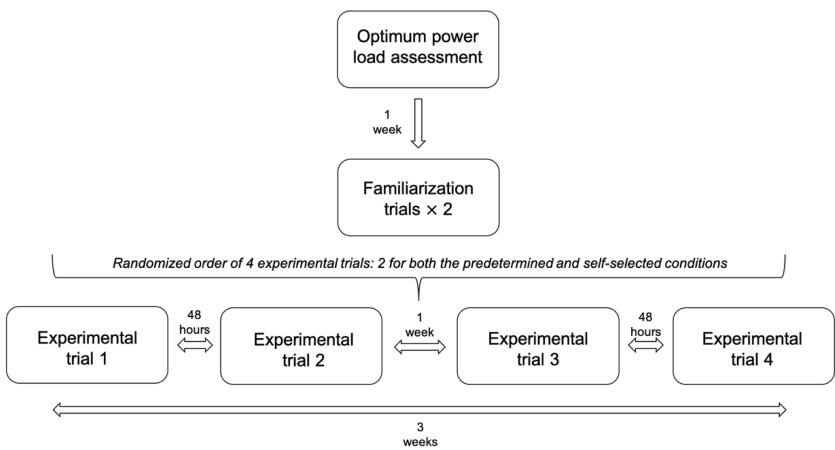

**Figure 1   Schematic representation of the study design.**

jump squat. Across sessions, the exercises were loaded with the same external resistance—optimum power load (OPL)—but structured with different exercise orders: (1) self-selected, in which participants chose their preferred order of exercises, (2) predetermined, in which the order was imposed on the participants. Power outputs were estimated using a linear position transducer. Subjective responses of fatigue and enjoyment were collected using the rate of fatigue (ROF; *Micklewright, Gibson & Salman, 2017*) and the Physical Activity Enjoyment Scale (PACES; *Teques et al., 2017*) questionnaires, respectively. Data was collected over a 3-week period in which participants completed one OPL assessment session, and four experimental sessions. Specifically, participants performed each of the experimental trials twice: two sessions with the self-selected exercise order, and two other sessions with the predetermined exercise order. The decision to adopt a repeated measures design in this study was two-fold. First, it allowed collecting multiple data points from each participant which increased statistical power. Second, the outputs of the two trials performed in each condition were compared to investigate the reliability of the measures across the same condition. The order of the experimental trials was counterbalanced and determined by block randomization. Following the OPL assessment, participants also completed one familiarization session for each condition to become acquainted with the experimental procedures. The purpose of the study was kept somewhat vague so as not to bias the results (*Halperin et al., 2017*). All sessions were completed in the same strength and conditioning training facility, separated by at least 48–72 h and supervised by two of the academy's strength and conditioning coaches and two researchers. Participants were instructed to avoid intense training 24 h prior to each testing day, refrain from caffeine ingestion for 24 h prior to testing and instructed not to eat for 2–3 h prior to each trial.

## Optimum power load assessment

In the first session, OPLs of the four exercises were determined for each participant. A linear position transducer (Chronojump, Barcelona, Spain) sampling at 1,000 Hz, fixed to the bar at a perpendicular angle to the floor, and the commercial software provided

**Table 2  Descriptive data of OPL and correspondent MPP and reliability (CV% and ICC) scores of MPV across the four sessions for each exercise.**

| Exercise | OPL (kg) | | MPV (m/s) | | |
|---|---|---|---|---|---|
| | Mean ± SD | Range | Mean ± SD | CV (95% CI) | ICC (95% CI) |
| Back squat | 37.0 ± 11.5 | 21–57 | 0.821 ± 0.082 | 2.84 (2.39, 3.29) | 0.92 (0.84, 0.97) |
| Jump squat | 32.8 ± 13.7 | 10–55 | 1.024 ± 0.088 | 2.18 (1.60, 2.76) | 0.87 (0.74, 0.95) |
| Bench press | 30.6 ± 7.9 | 30–43 | 0.827 ± 0.070 | 1.55 (1.24, 1.86) | 0.96 (0.92, 0.98) |
| Bench throw | 21.7–6.0 | 10–31 | 0.858 ± 0.067 | 1.94 (1.73, 2.16) | 0.93 (0.85, 0.97) |

Notes.

CV, coefficient of variation; 95% CI, 95% confidence interval; ICC, intraclass correlation coefficient; OPL, optimum power load; MPV, mean propulsive velocity.

by the manufacturer for the device, were used to measure the velocity of the bar during the four exercises. Criterion validity (i.e., comparison against a gold standard optical motion capture system) as well as high reliability scores for the velocity derivative measures collected with this device were recently reported by *Pèrez-Castilla et al. (2019)*. Participants performed a general warm up consisting of three sets of ten body-weight squats followed by five squat jumps or three sets of ten press ups followed by five plyometric press ups, before the lower body and upper body OLPs assessments, respectively. Then, warm up sets with progressively heavier loads were performed in each exercise. Standardized instructions were used for the exercises' execution (*Dello Iacono, Beato & Halperin, 2020a*). During the OPL assessment, each participant performed three repetitions at maximum velocity for each progressive load. A starting load corresponding to 40% of body mass, with 10 kg and 5 kg increments per set were used in the back squat and bench press exercises, respectively. A starting load corresponding to 20% of body mass, with 10 kg and 5 kg increments per set were used in the jump squat and bench throw exercises, respectively. Loads were progressively increased until a consistent decrement in velocity output was detected (*Loturco et al., 2015*). This equated to approximately five testing sets per exercise (range: 4–6). The OPLs were determined following the protocol described by *Loturco et al. (2015)* and were identified as the absolute load resulting in the highest mean propulsive velocity and the estimated corresponding power values observed during the trials. The individual OPLs in each exercise (Table 2) were then used for the two experimental protocols.

## Experimental protocols

The order of the four exercises in the two conditions was determined in the following manner. For the predetermined condition, the exercise order (back squat; jump squat; bench press; bench throw) was established by consensus between two accredited strength and conditioning coaches and two senior academic staff members in line with an evidenced based practice approach (*Ratamess et al., 2009*; *Sands, Wurth & Hewit, 2012*). For the self-selected condition, a month prior to the study commencement, each participant was asked to indicate the preferred order of the four exercises. Additionally, one week prior to the OPL assessment, each participant was given the choice to either confirm or change the previously selected exercise order. Only one participant changed the exercise order initially selected. Twelve different exercise orders were chosen by the participants (Fig. 2), and only

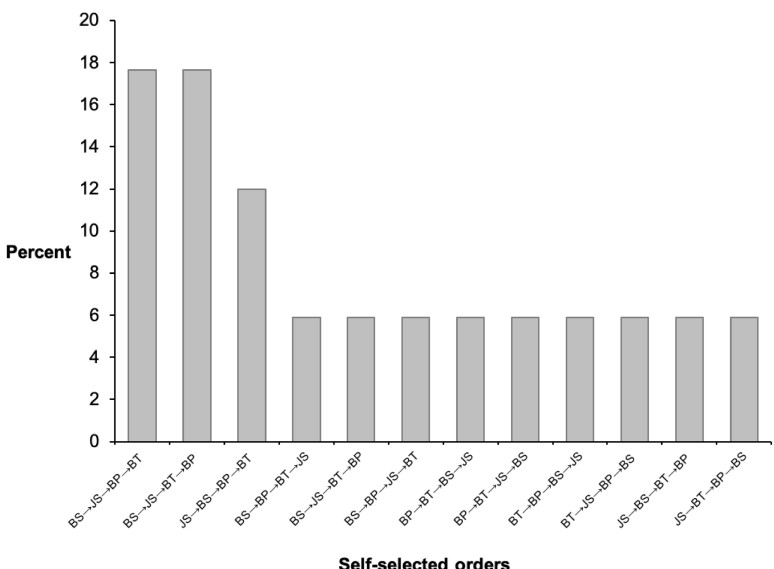

**Figure 2** **Exercises orders in the self-selected experimental condition.** BS, back squat; BP, bench press; BT, bench throw; JS, jump squat.

three participants selected the exact same order of the predetermined condition. In each condition, participants performed the same standardized warm up (see OPL assessment paragraph) followed by three sets of six repetitions for each exercise loaded with the OPL in either the self-selected or predetermined order, with two minutes rest between the sets and three minutes rest between the exercises. Participants were asked to move the bar as fast and as explosively as possible in all repetitions.

## Measures
### Power outputs
Mean propulsive power ($P_{mean}$) outputs were estimated from the mean propulsive velocity values monitored during the entire experimental trials with the same linear position transducer and commercial software described above. $P_{mean}$ outputs for each exercise were then expressed as relative to body mass (W/kg).

## Self-report measures
At baseline and immediately after each exercise, the 11-point ROF Likert scale (*Micklewright, Gibson & Salman, 2017*) was used to assess subjective perceived non-local muscular fatigue (*Othman et al., 2017*). The scale ranges from 0 ("not fatigued at all") to 10 ("total fatigue and exhaustion-nothing left"). The question "How fatigued are you at the moment?" was presented at the top of the scale. At the end of each protocol, participants also completed the PACES questionnaire to rate their level of enjoyment (*Teques et al., 2017*). Eight items were rated on a 7-point bipolar rating Likert scale which ranged from 1 ("I do not agree at all") to 7 ("I totally agree"). The single-item responses were tallied to calculate a summative score which was used for analysis. Higher PACES scores reflect greater levels of enjoyment.

## Statistical analyses

All data are presented as means ± standard deviation (SD) and confidence interval (95% CI). The Shapiro–Wilk test, with skewness and kurtosis values smaller than two (*Leech & Onwuegbuzie, 2002*), served to investigate the normality distribution of the absolute data. Reliability of the velocity outputs collected across the four sessions for each exercise was assessed using the coefficient of variation (CV; percent score [%]) and the intra-class correlation coefficient ($ICC_{3,1}$). High reliability was determined as a CV <5% and an ICC >0.70 (*Cormack et al., 2008*). We compared the effects between the two conditions on power outputs using a 2 (condition: predetermined/self-selected) × 4 (exercise: bench press/bench throw/back squat/squat jump) × 6 (repetition number: 1 to 6) repeated-measures Analysis of Variance (ANOVA). For this purpose, the $P_{mean}$ outputs were averaged across the same repetition number (e.g., 1st to 6th repetition) across the six sets (3 sets × 2 sessions) completed per exercise. The average ROF responses per exercise were calculated between the two sessions completed in each condition. Then, the effects on ROF responses between the two conditions were compared using a 2 (condition: predetermined, self-selected) × 4 (exercise: back squat, squat jump, bench press, bench throw) repeated-measures ANOVA. A paired-sample $t$-test was used to compare the effects on the PACES responses which were also averaged across the two sessions completed in each condition. The significance level was set at $p < 0.05$. Post hoc analyses were conducted using the Holm-Bonferroni correction for the $p$ values and CIs (*Dragicevic, 2016*). Finally, Cohen's $d$ (Mean difference/SD average) effect sizes (ES) were determined to provide qualitative descriptors of standardized effects and interpreted using the following criteria: trivial <0.2, small 0.2–0.5, moderate 0.5–0.8 (*Cohen, 1992*). All statistical analyses were conducted using Jamovi (version 1.2.9.0, Newcastle, UK).

We note that due to the fact that the reliability of the measures is a topic beyond the primary aim of this study, the summary of the supplementary reliability analysis and a brief discussion of the results are reported in the Data S1 file.

## RESULTS

All data were normally distributed. High reliability scores were observed consistently across all exercise with all CVs <5% and ICCs >0.70 cut-off thresholds (See Table 2). Individual data plots of power outputs for all thirty-six repetitions (2 sessions × 3 sets × 6 repetitions) performed during each exercise per condition are reported in Fig. 3. Average power outputs values (i.e., 36 repetitions) in each exercise and in both conditions are shown in Fig. 4. Descriptive, inferential and qualitative statistics of all variables across the two conditions are reported in Table 3. No statistical differences were observed between conditions ($F_{(1,16)} = 4.05$, $p = 0.06$) (See Fig. 4), interactions between conditions and exercise ($F_{(3,48)} = 0.16$, all $p \geq 0.18$), or between conditions and repetitions ($F_{(15,240)} = 0.44$, all $p \geq 0.08$; all ES $\leq 0.21$) on power outputs (See Fig. 3). Note that these findings were accompanied by trivial to small effect sizes (See Table 3). No between-condition statistical differences were found for the ROF responses in any of the exercises ($F_{(3,48)} = 0.96$, all $p \geq 0.41$; all ES $\leq 0.33$). Finally, the paired-sample $t$-test revealed no differences between the two conditions on the PACES responses ($p = 0.72$; ES $= 0.05$).

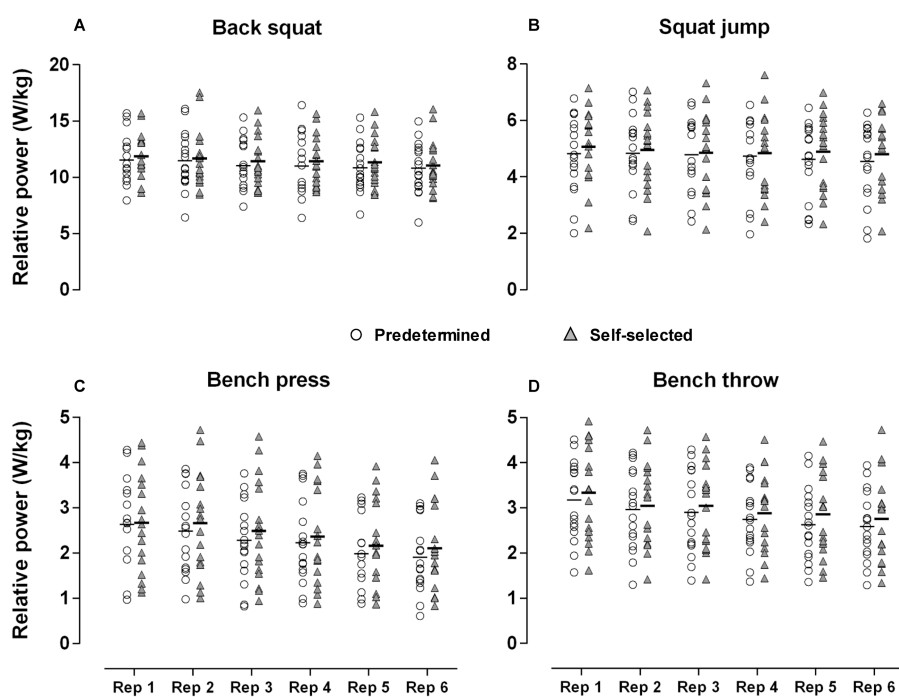

**Figure 3** **Individual data plots of power outputs for all repetitions performed during each exercise per condition.** Each data point indicates the average power output across the same repetition number (1st to 6th) performed in each exercise. (A) Back squat, (B) squat jump, (C) bench press, (D) bench throw.

**Table 3** **Descriptive (mean ± SD), inferential (95% CI) and qualitative statistics (ES) of all variables between self-selected and predetermined conditions.**

| Variable | Self-selected | Predetermined | Δ (95% CI) | ES |
|---|---|---|---|---|
| Back squat (W/kg) | 11.47 ± 2.12 | 11.11 ± 2.27 | 0.35(−0.56, 1.27) | 0.16 |
| Jump squat (W/kg) | 5.03 ± 1.64 | 4.72 ± 1.3 | 0.31(−0.27, 0.89) | 0.21 |
| Bench press (W/kg) | 2.41 ± 1.05 | 2.25 ± 0.84 | 0.16(−0.18, 0.50) | 0.17 |
| Bench throw (W/kg) | 2.98 ± 0.93 | 2.83 ± 0.83 | 0.15(−0.01, 0.32) | 0.17 |
| ROF Back squat | 5.62 ± 1.59 | 5.47 ± 1.157 | 0.15(−0.74, 1.03) | 0.09 |
| ROF Jump squat | 6.44 ± 1.37 | 6.53 ± 1.62 | −0.09(−0.63, 0.46) | −0.06 |
| ROF Bench press | 5.88 ± 1.27 | 6.35 ± 1.59 | −0.47(-1, 0.06) | −0.33 |
| ROF Bench throw | 6.12 ± 1.49 | 6.23 ± 1.68 | −0.12(−0.83, 0.59) | −0.07 |
| PACES | 38.3 ± 5.6 | 38.0 ± 7.2 | 0.32(−1.58, 2.23) | 0.05 |

**Notes.**

Note: descriptive data are reported as the mean ± SD of the thirty-six repetitions performed during each exercise or subjective questionnaire responses across the two sessions of each experimental condition.

Δ, mean difference; CI, confidence interval; ES, effect size; ROF, rate of fatigue; PACES, physical activity enjoyment scale; W, watt; kg, kilogram.

## DISCUSSION

We examined if providing choices over the order of exercises performed during strength and conditioning training sessions would enhance acute power performance and perceptions of fatigue and enjoyment as compared to a predetermined exercise order in elite youth

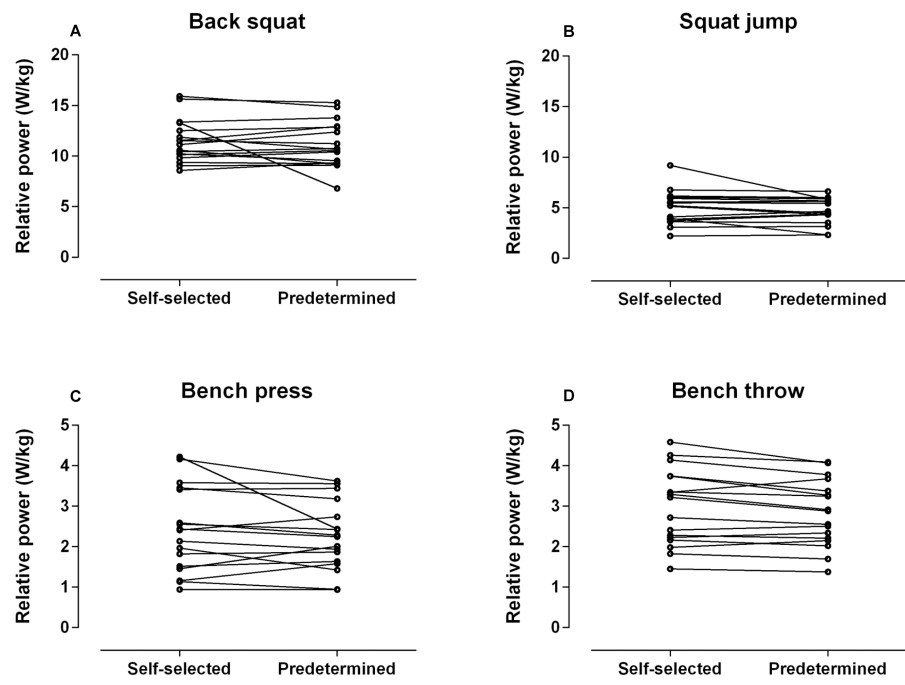

**Figure 4** **Illustrates the comparative scores between the self-selected and predetermined conditions on power output per exercise.** Each data point indicated the average power outputs values (i.e., 36 repetitions) for each participant. (A) Back squat, (B) squat jump, (C) bench press, (D) bench throw.

athletes. In contrast to our hypothesis, we found no statistical or practical differences in power outputs, ROF and PACES scores between conditions.

The main findings of this study are aligned with some (*Colquhoun et al., 2017*; *Emanuel et al., 2020*) but not most studies on this topic (*Balaguer et al., 2012*; *Gonzalez et al., 2016*; *Halperin et al., 2017*; *Kristiansen & Roberts, 2010*). For example, competitive boxers punched harder and faster when given a choice regarding the order of punches delivered in a combination, compared to a predetermined order (*Halperin et al., 2017*). Similarly, professional basketball players jumped higher after self-selecting the number of repetitions to complete in a post-activation performance enhancement protocol, compared to a fixed number of repetitions (*Dello Iacono, Beato & Halperin, 2020b*). The discrepancy between the results of the present study and those observing performance enhancing effects may stem from a number of reasons.

First, some studies were completed in controlled laboratory environments, whereas the current study was conducted in the athletes' natural training setting. Second, the environment of this study was highly representative of real-life, but it came at the expense of less control over various confounding variables (e.g., music, number of persons in the room, peer-to-peer feedback), which may have reduced the effects of choice provision (*Halperin, Pyne & Martin, 2015*). Third, the timing of choice provision in the current study was set weeks before it commenced for practical reasons. This is in contrast to most other studies in which choice was provided on the same day, usually prior to the motor task

completion (*Dello Iacono, Beato & Halperin, 2020b*; *Emanuel et al., 2020*; *Halperin et al., 2017*). As a consequence, the time delay between the final choice of exercise order during the self-selected condition and the event may have limited the positive mediating effects of the choice. However, this speculation remains to be investigated. Finally, the characteristics of the participants and the demands of their programs may have also played a role: young, elite level athletes, who are required to participate in the strength and conditioning sessions as part of their ongoing training routine. The very fact that the sessions represented part of the athlete's regular and mandatory training schedules could have also influenced the extent of the choice effects.

Given the similar power performance and psychological outcomes observed between the conditions, both can be implemented interchangeably or even concurrently in strength and conditioning sessions. This assumption is supported by the large range of exercise orders selected by the athletes (See Fig. 2). When the training goal in a given session is to optimize power outputs, our findings indicate that allowing athletes to choose the order of exercises can be effective in maintaining performances regardless of the order the athletes ended up with during the sessions. These findings also have practical importance. Given the trivial differences in power outputs, resistance training programs based on the self-selected approach have an advantage as they are simpler to plan and follow. This approach requires less constraints, facilitating session planning for the coaches and session execution for the athletes. Indeed, the likelihood of athletes waiting in line for equipment is minimized. Additionally, fostering independence among developmental youth athletes through choice provision over training variables may lead to positive psychological adaptions such as increased accountability and commitment to the session goals. However, we note that these results should be treated with cation as our study was conducted as an acute design and it is unclear if the same effects persist over time.

This study has a few limitations worthy of discussion. For practical reasons, the final sample size consisted only of seventeen subjects of which three completed two rather than four experimental sessions. To partly overcome the limitations associated with small sample sizes, we implemented a within-subject repeated-measures design including double sessions per condition which led to the collection of more than 2,000 repetitions that were subsequently analyzed. Participants were elite young team sport athletes, so whether these findings can be generalized to other individuals (e.g., adult athletes from either individual or team sports) require further research. Participants were granted with only one choice regarding one training variable (i.e., the order of exercises). This fact limits the generalizability of the results to other training scenarios including different exercises, their configuration, intensity and volume. Future studies are warranted to investigate whether granting athletes with other choices or combination of choices over a few training variables may lead to effects similar to those observed in this study. Finally, due to logistical constraints, participants had to pick and confirm their preferred order of exercises one month and one week before the study commenced, respectively. Therefore, it is worth investigating to what extent the timing of choice provision mediates the associated effects, and whether the latter can be optimized with preferences expressed more imminently to the performed tasks.

## CONCLUSIONS

We found that sessions in which the order of exercises was either predetermined or self-selected led to comparable power performances and psychological effects among youth athletes. Thus, self-selecting the exercise order is a valid alternative to a predetermined approach. These findings are of practical value as they suggest coaches adopting self-selected strategies to individualize prescription in resistance training practice based on individual preferences.

## ACKNOWLEDGEMENTS

We would like to thank the academy's Sport Performance Manager for granting access to the facilities and pupils, the Head Hockey Coach for his support in participant recruitment, the Assistant Head of Strength and Conditioning for his support in managing the experimental trials within the academy's S&C programme and the participants.

### Funding

The authors received no funding for this work.

### Competing Interests

The authors declare there are no competing interests.

### Author Contributions

- Kevin Watson conceived and designed the experiments, performed the experiments, authored or reviewed drafts of the paper, and approved the final draft.
- Israel Halperin conceived and designed the experiments, analyzed the data, prepared figures and/or tables, authored or reviewed drafts of the paper, and approved the final draft.
- Joan Aguilera-Castells performed the experiments, authored or reviewed drafts of the paper, and approved the final draft.
- Antonio Dello Iacono conceived and designed the experiments, performed the experiments, analyzed the data, prepared figures and/or tables, authored or reviewed drafts of the paper, and approved the final draft.

### Human Ethics

The following information was supplied relating to ethical approvals (i.e., approving body and any reference numbers):

The University of the West of Scotland approved this study (Ethical Application Ref: 2019-8652).

### Data Availability

Raw data, reliability analysis, a brief interpretation of the results are available in the Supplemental Files.

## Supplemental Information

Supplemental information for this article can be found online at http://dx.doi.org/10.7717/peerj.10361#supplemental-information.

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
