# Peer review of "A comparison between predetermined and self-selected approaches in resistance training: effects on power performance and psychological outcomes among elite youth athletes"

_PeerJ, doi:10.7717/peerj.10361_

## Round 0.1 · original submission · Major Revisions

Reviewers generally provided positive feedback regarding the manuscript. However, they request several changes that the authors should consider carefully.

·

Basic reporting

No comment

Experimental design

No comment

Validity of the findings

No comment

Additional comments

Comments to the Author
This study can be considered highly original due to the examination of the effects of choice provision over exercise order in strength training sessions on performance and subjective responses of fatigue. The article is well written, novel and with practical applications for S&C coaches who works with multiple players in the GYM at the same time. Thanks for give me the possibility to review this article and I have only few comments in order to help in the final version of the manuscript.

Material and methods
- Line 119, table 1. Please indicate (mm) in the sum of skinfolds. I recommend inform in the legend of the table what are the 8 skinfolds measured.
- Regarding the OPL assessment, could be interesting for the readers to show a table with the information of the load and the mean propulsive velocity at which the optimum power load was estimated for each exercises (mean, SD and range). In addition to this, could you inform about the ICC and CV inter-sessions per variable?
- Power outputs.- Please reflect that the power values were estimated using the linear transducer. The linear transducer does not measure force.
- A sample rate of 25 Hz (lineal encoder of 100 Hz) is more than adequate to record raw speed data during resistance training exercises (Bardella et al. 2016). Higher sampling frequencies provide no increase in the recording precision and may instead have adverse effects on the overall data quality. Why to use a lineal encoder with a sample rate of 1000 Hz during these exercises?

- Line 200. The athletes performed 3 sets of 6 rep for each exercises loaded with the OPL. How did you obtained the power values showed in the table 2? Is the best rep in mean power? The average of the 6 rep? the average of the 18 reps (3x6rep)? You have to explain this information in the methods. Sorry if I didn’t see this explanation.

Discussion
- Line 266 to 272. You don’t need to explain again this information in the discussion. You can describe in a short and precise way the aim of the study, and then the main findings of the investigation.
- Be careful when you are talking about performance in the article… performance is a very global concept. Finally you only measured power performance in 4 exercises, you should be more precise about this. Its also important that you didn’t evaluate longitudinal changes after a training process, therefore you are comparing if power performance within the training session was the same or different in both conditions.

Conclusions
- Don’t talk about the hypothesis again in the conclusions… Stay on topic. Reflect the conclusion of the study and practical applications.

·

Basic reporting

In my opinion, the manuscript seems well written. However, I cannot assess if the English in the present manuscript has the publication quality needed.
The introduction is well documented and written. However, to the best of my knowledge, no previous research was conducted with power training. Nunes (2020) and Simao (2012) reviews only included strength training focus on increasing maximal force and hypertrophy, excluding power training. If that is correct, the authors may include this literature gap in their introduction. On the other hand, if the authors know previous research that performs this type of comparison between self-selected and predefined order in power training, please include in the introduction.
In the discussion section, line 276, the authors have cited Halperin et al., 2015, but this paper doesn't deal with the topic under discussion. Maybe it is a mistake, and the authors want to cite Halperin et al., 2018.
In the same sentence, the authors have used four papers about autonomy from the self-determination framework. In my opinion, we cannot consider that this design with four sessions of training where the participants select the order only on two of them and without control of the rest of conventional training will influence the autonomy supportive.

Regarding tables:

- Please include a space between numbers and ± to improve the readability.
- The authors must use the same level of precision in the mean and the standard deviation.
- Table 1. The authors include the sum of 8 skinfolds as a descriptor of the participants. The authors must include the method of collecting these skinfolds and what the 8 skinfolds are in the method section and describe them in the legend of the table. The authors must also indicate the units of skinfolds on the table (i.e., mm).
- Table 2. The authors must indicate that the results for each exercise and condition are the mean of all repetitions.
- In my opinion, the authors may include a table with the individual data for each repetition and statistical analysis to improve the results section's clarity.
Regarding raw data, the authors only include the ID of the participants in the first sheet. Please indicate it on the other sheets, too.
Regarding supplemental Data, change "control condition" for predetermined order of exercises.

Experimental design

The research question is relevant, meaningful, and well defined. However, the fact that participants select the order of exercises weeks before the sessions is a significant limitation to answer the research question, but the authors describe it well in the discussion section.

Validity of the findings

Although the authors indicate that the participants select different exercise orders and that they had the chance to change the selection one week before the training, they don't specify if any participant changed his/her previous selection and the concrete order of exercises for each participant (at least on the raw data). This information could provide arguments to hypothesize about the validity of select exercise weeks before the training. The concrete order could allow us to identify if one exercise's order is different enough between both conditions.

Additional comments

The authors have used two references from the same authors and year (i.e., Dello lacono, A., Beato, M., & Halperin, I. (2020)). Please, use the letters “a” and “b” to identify them in the text.

Reviewer 3 ·

Basic reporting

The manuscript, for the most part, is written in clear and understandable English.
References and background detail has been provided, however, the final paragraph of the discussion appears to somewhat repeat previous points and does not add a great deal to the manuscript. I suggest that this section is re-evaluated by the authors.

Current figures are suitable but additional ones or information is required. See further comments to authors below. In addition, some additional information and reporting of data would have been useful to improve transparency.

In some sections it is difficult to follow the flow of the study and total number of completed experimental conditions. Suggest further clarity is required in text to avoid potential confusion for the reader. See also comments below.

Experimental design

The general rationale for the study is suitable.
Line 112-122: This section could be more specific and strengthened. For example, it is not clear what 'performance outcomes' refers to in this context. It should also be clearly stated that this is in the context of an acute training session and not over the course of a training intervention.

Some question, or lack of detail provided, regarding the determination of the load for the resistance exercises. This seems somewhat subjective and no report on whether the device used was reliable, or in fact, whether to variable of interest was reliable between trials in repetitions that used the same load. Suggest calculating this from the data or at the very least reporting reliability values of the device from other studies using similar exercises.

Clarity required about the number and order of experimental resistance training sessions. For example, the abstract states 5 experimental sessions, the methods states 'all four sessions', experimental design states 2 resistance training protocols, then line 195-197 seems to imply that there was only one-predetermined condition. Suggest Figure 1 is also presented in a way that more clearly shows the order of sessions.

The statistical approach on line 233-235 is difficult to follow. Please clarify exactly what was conducted here.

Validity of the findings

The authors report similar outcomes across all conditions. However, replicating this study may be difficult and some additional data would help with the transparency of the results and findings.

Some major points to consider:
1. It would be useful to report the load (kg) used for each exercise for each individual.
2. Would be useful to report the order of exercises in some fashion during the self-selected conditions. Perhaps showing this for each individual participant in a table then brief discussion on which was the most popular. This information could be used to develop pre-determined session structures moving forward as it is then informed by athlete preferences.
3. Perhaps the lack of differences between conditions can be somewhat explained by the fact that optimal power loads were used and so effort and fatigue would not be a severe as say a 'heavier' strength training session. If this was the case then more disparate effects between conditions may have been observed. This would make for an interesting point to expand upon in the discussion and highlight the rationale for further research across different types of resistance training sessions. Thus, any conclusions about the effectiveness of ether method is currently confined to this particular type of loading, population etc.
4. Data is collected for each repetition but no presented in this fashion. For example, the decline across reps and exercises may look visually appealing and provide further transparency.
5. Although not possible post-hoc I feel that assessing motivation prior to each resistance training intervention would have been of interest as this is also an important factor in terms of session performance and training adherence in non-supervised settings.
6. Some history on the training programs routinely undertaken by the athletes would be of use. For example, were the experimental conditions different to the exercises they usually perform? Do they normally conduct training using the pre-determined method? If so, the self-selection would be the novel condition but they would be familiarised already with the pre-determined method.

Additional comments

L83: 'can lead to greater'
L104-106: It is unclear what exactly is being referred to in this statement. I.e. how does it account for these factors?
L112: 'when planning'
L116: Suggest removing S&C and using full-term throughout, or at least in the first instance.
L166: Use full term for headings
L184: Please check year of the reference
L229: Was the data normal? It states that normality testing was conducted but the outcome is not presented.
L283: 'may stem...'
L286: Cleary state that the current study is being referred to here.
L315-316: 'our study was an acute design and it...'.
Supp File 1. Please check the reporting of Pearson's r. I think this should be reported as the 'r' value if the correlation is of interest, but 'r2' if the fit of the model is of interest.

---

## Round 0.2 · Minor Revisions

Reviewers are generally satisfied with the changes implemented in the manuscript. However, some additional changes are required.

·

Basic reporting

No comment

Experimental design

No comment

Validity of the findings

No comment

Additional comments

Congratulations

Reviewer 3 ·

Basic reporting

The authors have significantly amended the manuscript and as such this is a much improved version. The addition of extra tables and figures helps however, I still have some remaining comments that I have pasted in sections below.

Experimental design

The experimental design is sufficient and sections related to the description of the research questions and methodology have mostly been clarified. However, I have one remaining comment that I do not believe has been adequately addressed yet.

Validity of the findings

Impact and novelty of the study is appealing to the strength and conditioning community and this is now supported by extra data/figures and tables.
Conclusions are adequate but believe a previous comment still needs to be addressed to a higher standard.

Additional comments

In multiple sections the manuscript 'exercise order' now appears as 'exercises order'. However, in the format in most sections it should remain as 'exercise order' or alternatively the authors can say 'order of the exercises'.

L186-187: It is still not clear to the reader that there were four experimental sessions and two of each type when reading this statement. Please revise. Please also highlight somewhere why 2 of each session was conducted and the importance of this.

L242: 'corresponding'

Table 2: Please add unit of measurements for OP and MPV; perhaps kg and m/s?

I could not find anywhere captions for Figure 2, 3 and 4. Please provide.

Although I can appreciate the addition of the new figures in Figure 3 it is difficult to determine which column(s) correspond to which group. Suggest making the symbol a different shape for clarity.

It is still worth highlighting further that similar studies should be trialled with different types of resistance training paradigms (high load/volume etc). I acknowledge this was not part of the study but provides further direction for ongoing research. A line or two to state this would be sufficient.

---

## Round 0.3 · accepted · Accept

Congratulations for meeting the high standard publication of PeerJ.